

# High sensitivity Gd$^{3+}$- Gd$^{3+}$ EPR distance measurements that eliminate artefacts seen at short distances

Hassane EL Mkami[1], Robert I. Hunter[1], Paul A. S. Cruickshank[1], Michael J. Taylor[1], Janet E. Lovett[1], Akiva Feintuch[2], Mian Qi[3], Adelheid Godt[3], Graham M. Smith[1]

[1]SUPA, School of Physics and Astronomy, University of St Andrews, St Andrews KY16 9SS, UK.

[2]Department of Chemical Physics, Weizmann Institute of Science, Rehovot, Israel.
[3]Faculty of Chemistry and Center of Molecular Materials (CM$_2$), Bielefeld University, Universitätsstraße 25, 33615 Bielefeld, Germany.

*Correspondence to*: Hassane EL Mkami (hem2@st-andrews.co.uk) and Graham M. Smith (gms@st-andrews.ac.uk)

**Abstract.** Gadolinium complexes are attracting increasing attention as spin labels for EPR dipolar distance measurements in

biomolecules and particularly for in-cell measurements. It has been shown that flip-flop transitions within the central transition of the high spin Gd$^{3+}$ ion can introduce artefacts in dipolar distance measurements, particularly when measuring distances less than 3-4 nm. Previous work has shown some reduction of these artefacts through increasing the frequency separation between the two frequencies required for the Double Electron-Electron Resonance (DEER) experiment. Here we use a high power (1 kW), wideband, non-resonant, system operating at 94 GHz to evaluate DEER measurement protocols using two rigid Gd(III)-

rulers, consisting of two [Gd$^{III}$(PyMTA)] complexes, with separations of 2.1 nm and 6.0 nm, respectively. We show that by avoiding the $\left|-\frac{1}{2}\right\rangle \rightarrow \left|\frac{1}{2}\right\rangle$ central transition completely, and placing both the pump and the observer pulses on either side of the central transition, we can now observe apparently artefact-free spectra and narrow distance distributions, even for a Gd-Gd distance of 2.1 nm. Importantly we still maintain excellent signal-to-noise ratio and relatively high modulation depths. These results have implications for in-cell EPR measurements at naturally occurring biomolecule concentrations.


**Keywords**: Gadolinium (III), spin labels, ZFS, spin flip-flop, DEER






## 1 Introduction

DEER spectroscopy combined with Site Directed Spin Labelling (SDSL) is a powerful technique to probe structural and dynamic properties in a wide range of biological systems. Over the past decades, distance measurements have been mainly associated with nitroxide spin labels. This has led to the development of experimental protocols and reliable data analysis

programs for a routine extraction of distances and investigation of conformational changes. Amongst other reasons, the increasing interest in characterising proteins in their native environment has extended the spin labelling family to new labels based on paramagnetic metal ion complexes. $Gd^{3+}$-based spin labels have been of particular interest as they already exist as a major class of contrast agents used in MRI and show a strong stability against the oxidation or reduction conditions found in cells, making them an ideal candidate for in-cell distance measurements.

40         Gadolinium is a half integer high spin S=7/2 metal ion, characterised by a broad distribution of zero-field-splitting (ZFS) parameters and an isotropic g value at high field (Raitsimring et al., 2005). At lower temperatures its EPR spectrum is dominated by the central transition $\left|-\frac{1}{2}\right\rangle \rightarrow \left|\frac{1}{2}\right\rangle$ superimposed on a broad featureless background coming from all the other transitions. To first order, perturbation theory shows the central transition is independent of the ZFS interaction, while the other transitions scale linearly with the axial ZFS parameter $D$. However, to second order the central line narrows as the

operational frequency increases and its width scales proportionally with $\frac{D^2}{gB_0}$. Therefore, high frequencies have been favoured for distance measurements using $Gd^{3+}$-based spin labels due to an expected improved concentration sensitivity associated with placing the pump or observer frequency at the central line.

Since their introduction in 2007, several $Gd^{3+}$-based spin labels have been developed and a wide range of molecules have been successfully investigated, from simple model compounds to proteins, DNA, peptides and other biological systems

(Gordon-Grossman et al., 2011;Potapov et al., 2010;Raitsimring et al., 2007;Yagi et al., 2011;Shah et al., 2019). The good agreement between distance distributions derived from Gd-Gd DEER data and those resulting from other techniques has motivated researchers to attempt investigation of in-cell proteins and peptides (Qi et al., 2014;Yang et al., 2019;Dalaloyan et al., 2019). In most of these studies, $Gd^{3+}$ was treated like a S = 1/2 system and standard data analysis software packages, developed initially for nitroxides, have generally been applied. This approach has proven successful for Gd-Gd distances above

3-4 nm, but below 3-4 nm strongly damped dipolar distortions and artificially broadened distance distributions were obtained (Cohen et al., 2016;Dalaloyan et al., 2015;Feintuch et al., 2015;Manukovsky et al., 2017). This is caused by unwanted flip-flop transitions, whose effects are enhanced by strong dipolar coupling (Cohen et al., 2016;Manukovsky et al., 2017). This effect can be ameliorated by increasing the frequency offset between the pump and observer pulses (PO offset) (Cohen et al., 2016). This has usually been achieved by having the pump pulse positioned at the central transition and positioning the observer

pulse with as large a frequency offset as possible. This is usually difficult to achieve with standard resonator bandwidths on commercial instruments. Nevertheless, a high frequency dual mode cavity with an ingenious design has been demonstrated, which can accommodate pump and observer pulses with separations of more than 1 GHz (Cohen et al., 2016). Unfortunately such cavities, particularly at high fields and low temperatures, can be challenging to set up precisely. Relaxation-Induced



Dipolar Modulation Enhancement (RIDME) is another experimental alternative where no such restrictions apply, since it is a single frequency technique, however it suffers from overtones of dipolar frequencies and requires a more complicated data analysis (Collauto et al., 2016;Keller et al., 2017;Meyer and Schiemann, 2016).

In the present work, we demonstrate a simpler approach that uses a wideband non- or weakly-resonant sample-holder to show the benefit of wideband measurements. We use two Gd-rulers with calculated distances between the two [Gd$^{III}$(PyMTA)] complexes of 2.1 nm and 6.0 nm (Qi et al., 2016;Dalaloyan et al., 2015). The Gd-Gd distances were calculated for a temperature at 160.4 K, the glass transition temperature of the mixture of glycerol-d$_8$ and D$_2$O, 50/50 (v/v), applying the wormlike chain model. We explore two different experimental protocols. The standard approach where the pump pulse is positioned at the central transition, but with variable offset between pump and observer pulses of up to 900 MHz. In general, we observe narrower distance distributions and improved Pake patterns as frequency separation is increased. In the second approach we place the pump and observer pulses on either side of the $\left| -\frac{1}{2} \right\rangle \rightarrow \left| \frac{1}{2} \right\rangle$ central transition, avoiding excitation of the central transition altogether. In this case, we observe almost perfect Pake patterns, consistent with elimination of the artificial broadening of the distance distribution, even for Gd ruler (2.1 nm).

For this short 2.1 nm distance we show that any loss of sensitivity from not exciting the central transition is offset by the shorter time window now required to make the measurement.

## 2 Experimental

### 2.1 Sample preparation

The synthesis of the Gd-rulers has been described elsewhere (Qi et al., 2016). Solutions of 40 μM concentration (molecules) of Gd-ruler (2.1 nm) and Gd-ruler (6.0 nm) were prepared in 50/50 (v/v) deuterated glycerol and D$_2$O (for chemical structure see Fig. 1). The use of the glycerol-d$_8$ /D$_2$O (1:1, v/v) was dictated by the desire to obtain a good glass, to reduce scattering losses, and to extend the phase memory time. For the Q-band measurements the samples were transferred to standard 3 mm quartz EPR tubes and flash frozen in liquid nitrogen prior to loading into the spectrometer. For the W-band experiments, the samples were transferred into 27 mm long fluorinated ethylene propylene (FEP) tubes with 3 mm outer diameter and 2 mm inner diameter and flash frozen in liquid nitrogen prior loading into sample-holder cartridges that were separately precooled in liquid nitrogen. These sample-holder cartridges were then loaded into the W-band spectrometer which had been pre-cooled to 150 K.



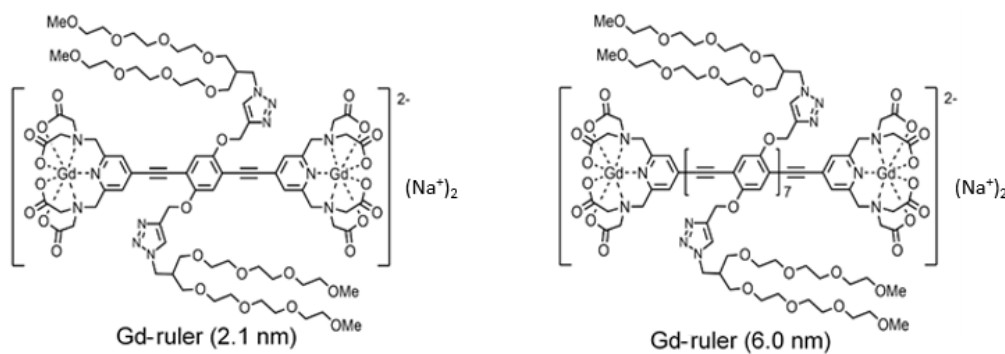

**Figure 1**: Chemical structures of Gd-rulers used in the current study.

**2.2 Spectrometers**

The spectrometers used for these measurements were a Bruker ELEXSYS E580 high power (150 W) Q-band pulsed spectrometer with ER 5106QT-2W resonator, and a home-built 1 kW W-band spectrometer (93.5 - 94.5 GHz). This W-band spectrometer, widely known as HiPER, has been described previously (Cruickshank et al., 2009;Motion et al., 2017). It operates with a wideband non-resonant (or weakly-resonant) induction mode sample holder, which is now described in more detail.


**2.2.1 Non-resonant induction mode sample-holder**

A non-resonant induction mode sample-holder is essentially a shorted symmetrical waveguide where two dominant orthogonal linearly polarised modes can propagate. The incident linearly polarised beam can be decomposed into two orthogonal circular polarisation components. At resonance, one circular component is absorbed (or emitted) by the sample, resulting in a reflected

beam containing a cross-polarised component perpendicular to the linear polarisation of the incident beam. In the system described here these reflected signals are diplexed via a wire grid polariser to separate the high power incident beam from the very much smaller cross-polarised component which is passed to the detection system. The dimensions of the empty waveguide sample-holder (3 mm diameter) are chosen to be single-mode at the operating frequency of the spectrometer. The advantage of these shorted waveguide sample-holders are that they are inherently wideband since they are only weakly resonant. It might

be thought that sensitivity would be strongly diminished as there is no resonant enhancement of either the excitation pulse or signal. However, the critical parameter that determines the resonant enhancement is the microwave conversion efficiency of the sample holder, $c$, with units $G/W^{1/2}$ (Smith et al., 2008). A single-mode shorted waveguide at 94 GHz can have a comparable or better conversion efficiency (5-6 ns $\pi/2$ pulse with 1 kW input) than a commercial X-band pulsed resonator. Compared to a dual-mode W-band resonator, a shorted waveguide can also offer a hugely increased sample volume (up to 75 μL in the

current system), provided that sample dielectric losses are low, which is usually the case for measurements made at cryogenic temperatures. The non-resonant cavity also offers considerable flexibility in optimising excitation frequencies and bandwidths



at both pump and observer frequencies. Therefore these types of systems can have extremely high concentration sensitivity whilst offering very large instantaneous bandwidths. This potentially makes them ideal for Gd spin label distance measurements, especially when large separations between the observer and pump pulses are required.

120       The sample is placed in a FEP tube within a sample-holder cartridge and mounted into a spring-loaded mount (see Fig. S1a). The latter co-locates to a smooth cylindrical waveguide transmission line of diameter 3 mm, which supports two orthogonal $TE_{11}$ modes. Radiation is fed to the shorted waveguide via an adapted corrugated feedhorn that feeds to a wider bore corrugated pipe which in turn feeds to an optical system. An adjustable backshort consisting of a roof mirror with a shallow roof angle, is placed below the sample and its position can be adjusted vertically and rotationally by means of piezo-

motors (Attocube Systems AG). The optimisation and isolation of the cross-polarised signal component are crucial both during the experimental set-up and measurements and this is mostly achieved by fine adjustment of the adjustable backshort using the Attocubes and of the quasi-optical components.

      To facilitate the loading of pre-frozen samples, the samples and sample-holder cartridges are pre-cooled externally to the spectrometer in liquid nitrogen. The spring-loaded mount, feedhorn and corrugated pipe are housed inside a vacuum vessel

which forms an extension to the sample flow cryostat and includes a vacuum window at the top to allow access for the microwave beams. The sample cryostat is cooled until the temperature of the spring-loaded mount reaches 150 K which has been found to be a reliable temperature to use for loading of pre-frozen samples. In order to load the sample the flow cryostat must be stopped and returned to ambient pressure using helium gas. The vacuum vessel is hoisted up along with the corrugated pipe, feedhorn and spring-loaded mount whilst a continuous flow of helium gas is maintained to minimise icing of the cryostat

and microwave feed system. The sample-holder cartridge is removed from the liquid nitrogen and inserted into the spring-loaded mount along a guide channel where it becomes located into sockets, ensuring accurate alignment of the waveguide. The vacuum vessel is then lowered back down and sealed to the cryostat and is then evacuated. Cryogen flow is reinstated in the cryostat and the system is cooled to the measurement temperature.

**2.2.2 Frequency dependent power variation in HiPER**

It should be noted that the transmitted power level (from the amplifier / isolator / switch combination) is not constant over the whole range of the frequency offsets used in this study. This is illustrated in Fig. S1b where we show the power level versus frequency monitored at different points along the transmitter chain of HiPER. As a consequence of this, absolute modulation depths should not be compared quantitatively.


**2.3 Pulse EPR experiments**

For the Q-band experiments, echo detected field sweep (ED-FS) measurements were carried out at 10 K. The $\pi/2$ and $\pi$ pulse lengths were set at 16 and 32 ns respectively, with an inter-pulse delay of $\tau = 200$ ns.

For the W-band experiments all measurements were performed at 10 K, which corresponds to the optimum temperature for

these experiments when the central transition is excited (Feintuch et al., 2015;Goldfarb, 2014;Raitsimring et al., 2013). The



ED-FS spectra were recorded using a Hahn echo sequence with pulse lengths 6 and 12 ns as $\pi/2$ and $\pi$ respectively and a delay of 300 ns. These pulse lengths were optimised by setting the magnetic field to the peak of the spectrum. The echo decay ($T_m$) and the inversion recovery ($T_1$) experiments were recorded at the central maximum of the ED-FS spectrum by stepping the associated sequences with steps of 100 ns and 1 μs respectively. The repetition rate for all W-band experiments was set at 3

kHz, unless otherwise stated, and this was again optimised at the maximum of the ED-FS spectrum.

The DEER experiments were carried out using the standard dead-time free four-pulse sequence (Pannier et al., 2000).

$$\frac{\pi}{2}(obs) \rightarrow \tau_1 \rightarrow \pi(obs) \rightarrow t \rightarrow \pi(pump) \rightarrow \tau_1 + \tau_2 - t \rightarrow \pi(obs) \rightarrow \tau_2 \rightarrow echo$$

The echo intensity was monitored as a function of $t$. For Gd-ruler (6.0 nm), the pump pulse was applied, for all experiments, at the maximum of the ED-FS spectrum whereas the observer frequency was set at 94 GHz with different offsets from the pump frequency as reported in Table-1. For Gd-ruler (2.1 nm), different settings were investigated, with either the pump

160     frequency being set at the maximum of the ED-FS spectrum and the observer frequency placed on the side of the central line, or with both the pump and observer frequencies being set on either side of the central line. The experimental parameters used in both cases are summarised in Table-1. Optimisation of the observer and pump pulse lengths was carried out systematically for each experiment given the wide range of frequency offsets used in this study. It is necessary to re-optimise the pulse lengths for each frequency offset, due to power variation from the transmitter chain.

DEER data were processed using the DeerAnalysis (2019) program that allows extraction of distance distributions (Jeschke et al., 2006). Fits to the data were based on standard Tikhonov regularisation analysis using the bending point in the L-curve. The excitation profiles of the pump and observer pulses were simulated using a home written MATLAB-based program. The simulated ED-FS spectra and the associated sub-spectra for each transition were performed using the EasySpin

170     program (Stoll and Schweiger, 2006).





| Offset[1] (MHz) | Obs[2] $\pi$(ns) | Pump[3] $\pi$(ns) | $\tau_2$($\mu$s) | Data points | SRT[4] (kHz) | $\lambda$ (%) | Echo SNR | Time averaging | Number of averages[5] | Sensitivity measure[6] |
|---|---|---|---|---|---|---|---|---|---|---|
| *Pumping on the central line and observing on the side* | | | | | | | | | | |
| **Gd-ruler (6.0 nm)** | | | | | | | | | | |
| 120 (P$_1$O$_1$) | 11 | 10 | 10.3 | 251 | 3 | 6.0 | 1111 | 1h30min (14 scans) | 42000 | 2.06 |
| 120 (P$_2$O$_2$) | 16 | 16 | 10.3 | 251 | 3 | 5.0 | 769 | 44min (7 scans) | 21000 | 1.68 |
| 420 (P$_3$O$_3$) | 11 | 10 | 10.3 | 251 | 3 | 4.5 | 2000 | 10h20min (99 scans) | 297000 | 1.02 |
| **Gd-ruler (2.1 nm)** | | | | | | | | | | |
| 120 (P$_1$O$_1$) | 24 | 24 | 2.2 | 251 | 3 | 2.5 | 1923 | 1h (10 scans) | 30000 | 1.75 |
| 420 (P$_2$O$_2$) | 12 | 12 | 2.8 | 251 | 3 | 4.0 | 3125 | 1h (10 scans) | 30000 | 4.56 |
| 840 (PO$_5$) | 12 | 12 | 1.5 | 121 | 1 | 2.9 | 3279 | 1h40min (33 scans) | 33000 | 4.68 |
| 900 (PO$_6$) | 12 | 12 | 1.5 | 121 | 1 | 3.2 | 3846 | 0h45min (15 scans) | 15000 | 8.99 |
| *Pumping and observing on the sides of the central line* | | | | | | | | | | |
| **Gd-ruler (2.1 nm)** | | | | | | | | | | |
| 800 (P$_3$O) | 8 | 8 | 1.5 | 121 | 1 | 2.1 | 8333 | 1h26min (28 scans) | 28000 | 9.35 |
| 900 (P$_4$O) | 12 | 12 | 1.5 | 121 | 1 | 1.1 | 10000 | 4h27min (71 scans) | 71000 | 3.69 |

175  **Table-1**: Experimental settings parameters used for DEER measurements on both rulers and the associated modulation depths obtained by fitting the DEER data with DeerAnalysis (Jeschke et al., 2006). To allow different DEER measurements to be compared more easily we take our sensitivity measure as the echo SNR multiplied by the modulation depth divided by the square root of the total number of measurements. It should be noted that this does not take into account differences in excitation bandwidth of pump and observer pulses.

180  [1] Frequency separation between pump pulse set at position i (P$_i$) and observer pulse at position j (O$_j$).
[2,3] Observer and pump $\pi$ pulse lengths. The observer $\pi$/2 pulse was always half the observer $\pi$ pulse.



---

[4] This is the Shot Repetition Time.

[5] Number of averages calculated as: number of scans * number of shots per point.

[6] The sensitivity measure is calculated as $=\frac{\lambda * SNR(Echo)}{(\sqrt{total\ number\ of\ points\ measured})}$ where SNR (echo) is the ratio of the maximum echo height to the standard deviation of the noise. This is obtained by subtracting a smoothed fit from the data and then calculating the standard deviation from the resulting noise trace. The total number of points measured is the total number of averages per point multiplied by the number of points in a scan.

## 3 Results

### 3.1 EPR spectra and relaxation times

The ED-FS spectra for both samples are similar to those reported for other $Gd^{3+}$ complexes with a characteristic sharp line corresponding to the central transition and a broad featureless background resulting from contributions of all other transitions. The spectra recorded at Q- and W-bands are shown in Fig. 2. The simulation of the sub-spectra was performed using EasySpin by considering a distribution of the ZFS parameters $D$ and $E$ (Stoll and Schweiger, 2006). The magnitude and distribution of the ZFS depend primarily on the nature of the interactions between the $Gd^{3+}$ ion and the ligand and / or solvent molecules coordinating to the $Gd^{3+}$ ion. These are taken into account by the $D$ and E strain parameters used by EasySpin and they are defined as Gaussian distributions. Furthermore, it was shown in some cases that a bimodal Gaussian distribution centred on $D$ and $-D$ considerably improved the simulation (Raitsimring et al., 2005;Clayton et al., 2018). The $D$ parameters used for the simulations are reported with those obtained for other $Gd^{3+}$ complexes in Table-S1.

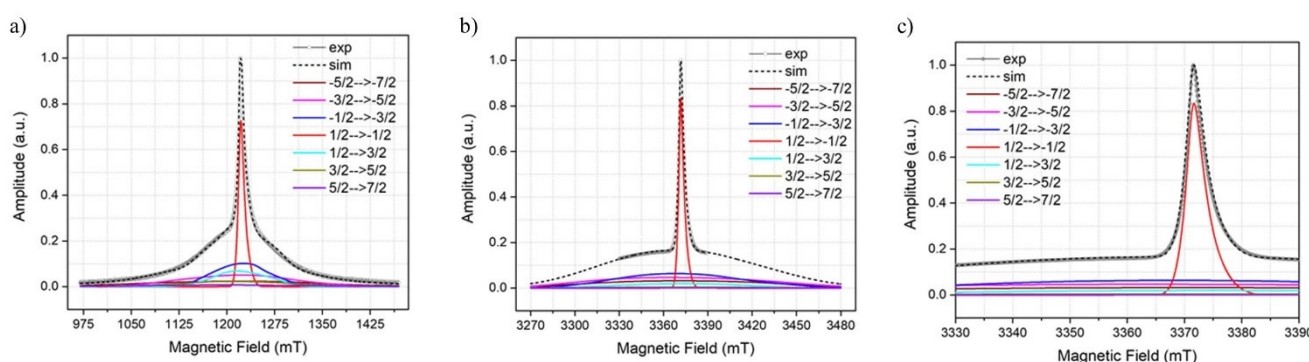

**Figure 2**: Simulated and experimental ED-FS spectra of Gd-ruler (6.0 nm) with the associated sub-spectra of the individual transitions, a) at Q-band, b) at W-band with wide magnetic field ranges and c) at W-band with narrow magnetic field ranges respectively.

The phase memory time, $T_m$, and the spin lattice relaxation time, $T_1$, were measured for both samples at 10 K with the magnetic field set on the central maximum of the ED-FS spectrum. In neither case was it possible to fit the data with a single exponential function. This finding has also been reported in other studies and seems to be typical for $Gd^{3+}$ complexes measured at low temperatures (Collauto et al., 2016;Raitsimring et al., 2014). The $T_m$ time traces were therefore fitted with a sum of two



stretched exponentials with fixed exponent values of 1 and 2 and the results are shown in Fig. S2a. $T_m$ values estimated from these fits are shown in Table-S2 and indicated fast and slow relaxation contributions to the echo decay. $T_1$ time traces were well fitted with a bi-exponential function as shown in Fig. S2b. Fast and slow time constants were derived from these fits and are reported in Table-S3.

### 3.2 Results from DEER spectroscopy

### 3.2.1 Gd-ruler (6.0 nm)

Background corrected DEER data obtained with Gd-ruler (6.0 nm) are shown in Fig. 3a and the corresponding primary data are shown in Fig. S3a. The pump and observer positions with their associated excitation profiles are reported in Fig. 3b, 3c and 3d. In addition to the excitation profiles, the pump and observer pulses positions, with respect to the central transition $\left|-\frac{1}{2}\right\rangle \to \left|\frac{1}{2}\right\rangle$, are shown in Fig. S4. The modulation depths derived from the fits are summarised in Table-1. The modulation depth λ of 6% obtained from the DEER data recorded with 120 MHz PO offset (see Table-1) is in a good agreement with that derived from the concentration dependence of Gd-4-iodo-PyMTA, the parent $Gd^{3+}$ complex of the ruler used here (Dalaloyan et al., 2015). By being slightly more selective with the pump pulse but keeping the same PO offset of 120 MHz, the modulation depth decreases to 5% as expected due to fewer spins being excited. When the PO offset is increased to 420 MHz the modulation depth drops to 4.5% mainly due to the output power drop off towards the band edges in our system. For this latter measurement, it should be noted the field was different than in the former two experiments, and the pump pulse has a different frequency (see Fig. 3b, 3c and 3d).





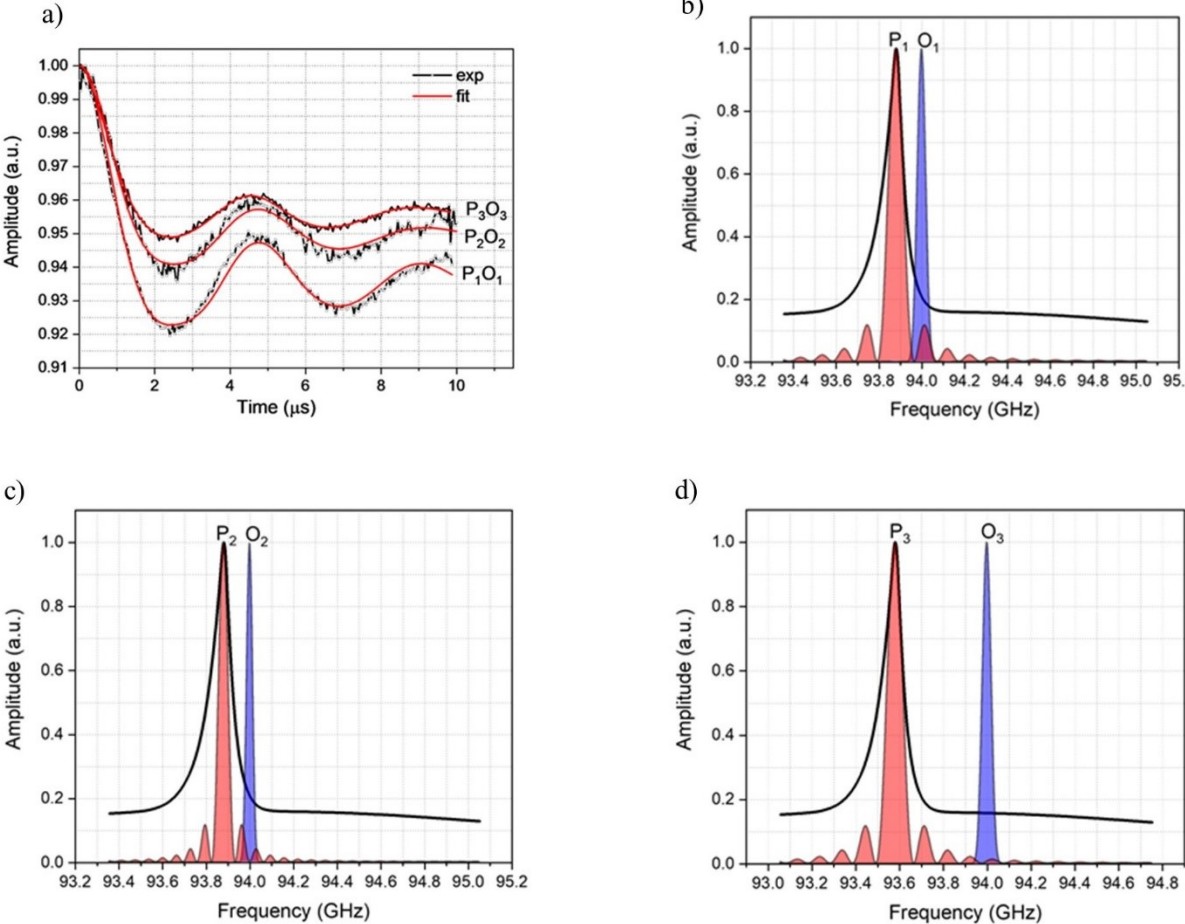

**Figure 3**: a) Background corrected DEER data (black curves) of Gd-ruler (6.0 nm) recorded with different PO offsets with the fits (red) obtained by DeerAnalysis (Jeschke et al., 2006). Excitation profiles of the pump (P) and observer (O) with PO offsets of (b,c) 120 MHz and (d) 420 MHz. Details of pump (P) and observer (O) pulses are summarised in Table 1.


The derived Pake pattern spectra and the associated distance distributions are shown in Fig. 4. The distance distribution derived from DEER data measured with 120 MHz PO offset appears to be deviating slightly from 6.0 nm, the expected distance for this Gd-ruler, with a full width half height (FWHH) of 0.56 nm whereas with 420 MHz offset it is well centred on 6.0 nm with a FWHH of 0.48 nm. The Pake patterns, for all experimental settings, show normal and typical shapes

with clear dipolar singularities corresponding to parallel and perpendicular orientations. These DEER measurements were recorded, as mentioned, with the pump position set at the peak of the FS-ED spectrum, which primarily excites the $\left|-\frac{1}{2}\right\rangle \rightarrow \left|\frac{1}{2}\right\rangle$ transition whereas the observer frequency for both offsets was positioned where the $\left|-\frac{3}{2}\right\rangle \rightarrow \left|-\frac{1}{2}\right\rangle$ transition contributes most





to the detected signal (see Fig. S4a,b). The Gd$^{3+}$ spectrum is the result of a superposition of several transitions with different weights, and their contributions, either to the pumped or observed spins, are expected to be magnetic field dependent. By

increasing the PO offset from 120 MHz up to 900 MHz, while keeping the pump position at the maximum of the FS-ED spectrum, the contribution of the $\left|-\frac{3}{2}\right\rangle \rightarrow \left|-\frac{1}{2}\right\rangle$ transition to the detected signal gradually decreases whilst the contributions of the other transitions, $\left|-\frac{7}{2}\right\rangle \rightarrow \left|-\frac{5}{2}\right\rangle$, $\left|-\frac{5}{2}\right\rangle \rightarrow \left|-\frac{3}{2}\right\rangle$, increase.

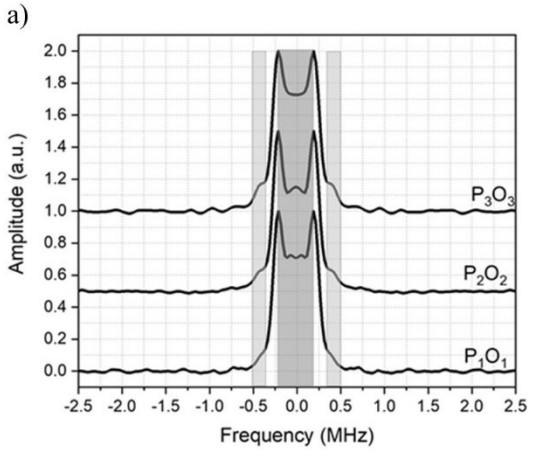
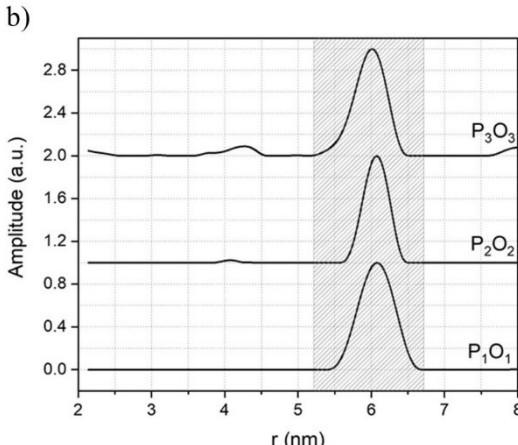

**Figure 4**: a) Pake Pattern spectra obtained from fitting of the DEER data of Gd-ruler (6.0 nm) measured with different offsets between pump and observer pulses and b) corresponding distance distributions derived from the DEER data. The positions of

the pump (P) and observer (O) are shown in Fig. 3b, 3c and 3d.

### 3.2.2 Gd-ruler (2.1 nm)

The DEER measurements with Gd-ruler (2.1 nm) were conducted with a combination of different pump and observer positions and several PO offsets. Figure 5a shows background corrected DEER data obtained with measurements performed with the

pump pulse set at the position of the central transition and PO offsets of 120, 420, 840 and 900 MHz. The corresponding primary DEER data are shown in Fig. S3b. The excitation profiles of the pump and the observer pulses at these positions are reported in Fig. 5b, 5c and 5d. The pump and observer positions with respect to the central transition are shown in Fig. S5. At 120 MHz and 420 MHz PO offsets, the time domain DEER data show severely damped dipolar modulations (see Fig. 5a) whereas in the cases of 840 and 900 MHz offsets, the dipolar modulations are significantly recovered, however they are still

not as well defined as one might expect for a stiff model system. The obtained modulation depths λ are reported in Table-1.



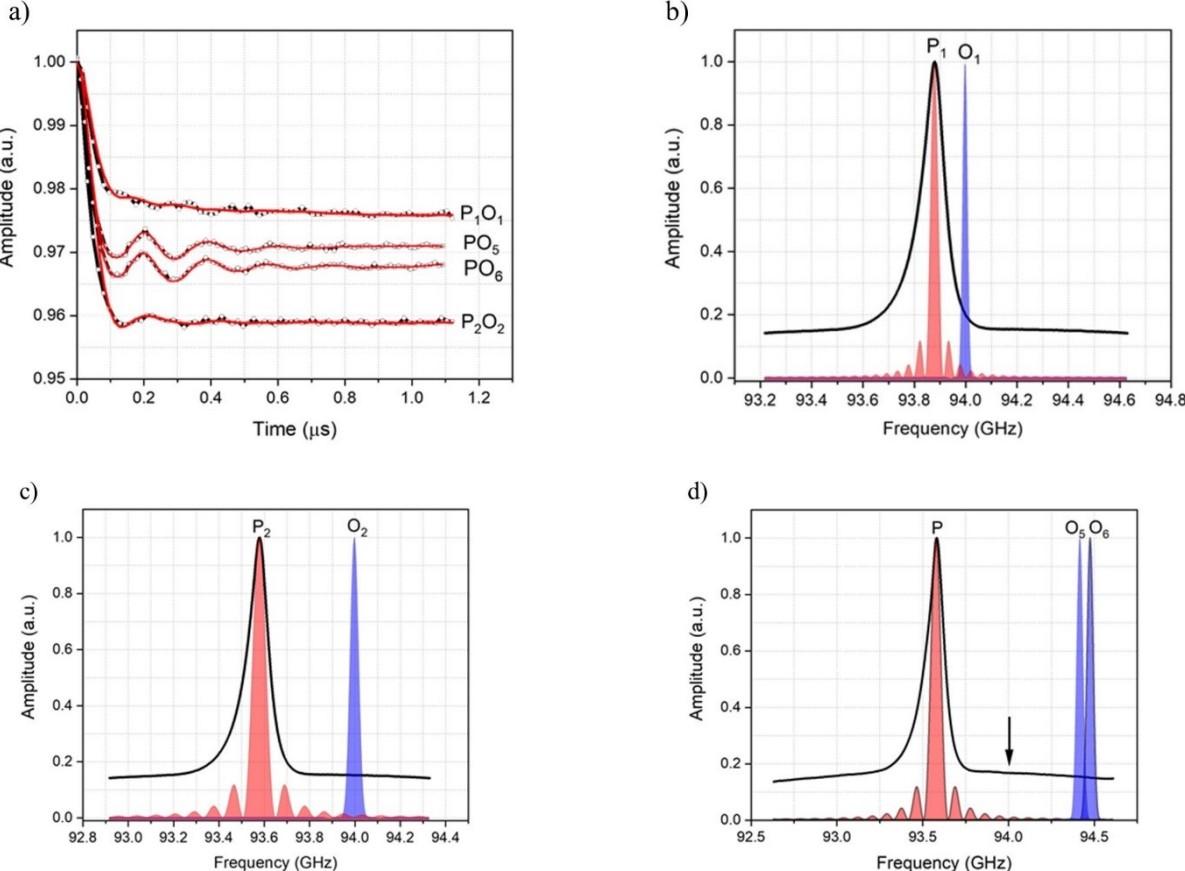

**Figure 5**: a) Background corrected DEER data (black curves) of Gd-ruler (2.1 nm) recorded with different offsets between pump and observer pulses together with fits (red curves) obtained by DeerAnalysis (Jeschke et al., 2006). Excitation profiles of the pump (P) and observer (O) pulses with PO offsets of (b) 120 MHz, (c) 420 MHz and (d) 840 and 900 MHz. Please note the different frequency scales. Details of the pump (P) and observer (O) pulses are summarised in Table-1.


We do not expect any orientation selection and so the Pake pattern spectra reported in Fig. 6a show strong distortions and poorly resolved dipolar singularity points for the 120 and 420 MHz PO offsets. In contrast, we observe substantially improved Pake pattern spectra for the larger offsets of 840 and 900 MHz, particularly concerning the perpendicular dipolar singularities. In Fig. 6b, the distance distributions are considerably broadened for the 120 and 420 MHz PO offsets, with

FWHH, determined only for the major peak centred at 2.1 nm, of 0.83 and 0.45 nm. However, at 840 MHz PO offset, the peak in the distance distribution is centred at the expected 2.1 nm distance with a FWHH of 0.24 nm. The best results were obtained with the 900 MHz PO offset giving a distance distribution with a FWHH of only 0.17 nm.



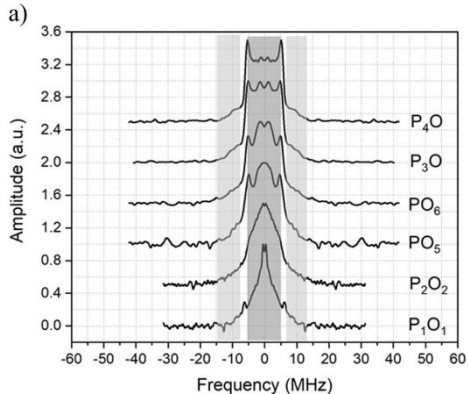
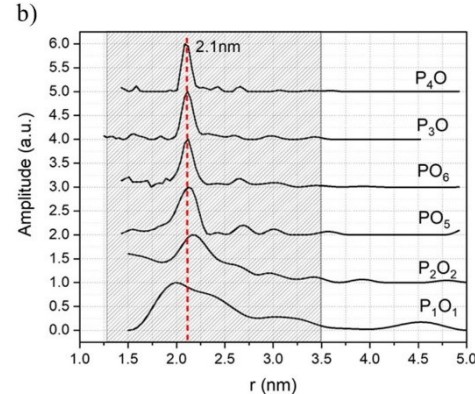

**Figure 6**: a) Pake pattern spectra obtained from the fitting of the DEER data of Gd-ruler (2.1 nm) measured with different

offsets between pump and observer pulses and b) associated distance distributions derived from the DEER data. The

corresponding positions of the pump (P) and observer (O) pulses are shown in Fig. 5b, 5c and 5d.

A further set of DEER experiments were performed by setting the pump and observer pulses on either side of the

central transition with large PO offsets. With this we aimed to exclude completely the contribution of the $\left|-\frac{1}{2}\right\rangle \rightarrow \left|\frac{1}{2}\right\rangle$ transition

from both the pumped and observed spins (see Fig. S6). Figure 7a shows the DEER data corresponding to 800 and 900 MHz

frequency PO offsets. The pulse profiles associated with the pump and observer pulses are presented in Fig. 7b. For both PO

offsets the obtained dipolar modulations show more than four clear oscillations and smooth damping, highly reminiscent of

spectra of structurally related nitroxide-rulers with similar distances (Godt et al., 2006;Jeschke et al., 2010).

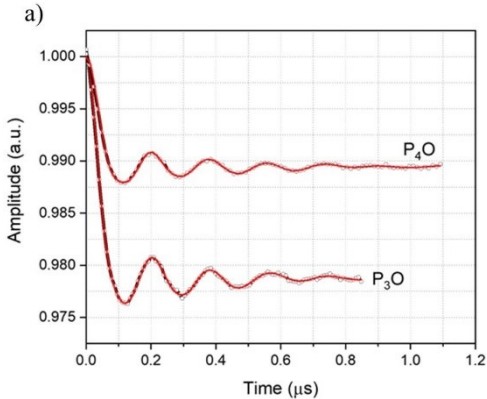
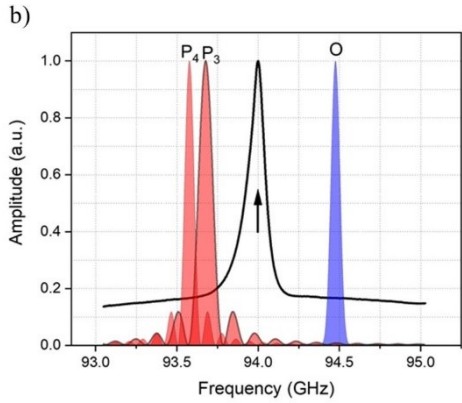

**Figure 7**: a) Background corrected DEER data (black curves) of Gd-ruler (2.1) nm recorded with different offsets between

pump and observer pulses together with fits (red curves) obtained by DeerAnalysis (Jeschke et al., 2006). b) Excitation profiles

of the pump (P) and observer (O) pulses at 800 MHz and 900 MHz frequency offsets. The pump (P) and observer (O) pulses

are summarised in Table-1.



The Pake patterns reported in Fig. 6a show the expected shape with well resolved perpendicular and parallel dipolar
singularities. The corresponding distance distributions, shown in Fig. 6b, show a very narrow major peak centred at 2.1 nm
with FWHH of 0.17 and 0.11 nm respectively.

## 4 Discussion

High quality DEER spectra from 40 μM samples were obtained with averaging times of an hour or two. Modulation depth,
SNR of the echo and experimental parameters are given in Table-1. As different traces were measured with different numbers
of scans and different shot repetition times, or have different numbers of points in the scan, we also provide a sensitivity
measure for DEER measurements that normalises for these quantities. Results can be compared to W-band measurements on
the same Gd-rulers (Dalaloyan et al., 2015). The high concentration sensitivity, relative to conventional W-band resonator-
based spectrometers is attributed to much larger effective sample volumes (~ 50 - 80 μL) and larger excitation bandwidths
facilitated by the available high power that is only partially offset by the lower conversion factor. Large effective volumes are
possible with biological systems in non-resonant sample-holders at W-band, as dielectric losses are expected to be small at
cryogenic temperatures (tan δ < 0.001) if a high quality glass is formed.

The best fit to the echo decays (measured at the central transition), shown in Fig. S2a, were obtained with two
stretched exponential functions with exponents being fixed to 1 and 2. Little difference in phase memory time was observed
between the two rulers with a slightly lower decay for Gd-ruler (2.1 nm). However, no correlation had been found between the
echo decay rate and the Gd-Gd distance for the same type of rulers as used in our study (Dalaloyan et al., 2015). The deviation
from the mono-exponential appears to be a characteristic of the $Gd^{3+}$ complexes, and very similar results have been obtained
before (Collauto et al., 2016;Raitsimring et al., 2014;Dalaloyan et al., 2015). Nuclear spin diffusion is often the dominant
process in phase relaxation of the central transition (Garbuio et al., 2015), when one would expect the data to be well fitted
with a single stretched exponential with an exponent of close to 2 (Kathirvelu et al., 2009). However nuclear spin diffusion is
expected to be significantly reduced by matrix deuteration, and contributions resulting from thermally assisted fluctuations in
the zero-field splitting are then expected to become significant (Raitsimring et al., 2014). The need to fit with two stretched
exponentials suggests an additional dephasing process is contributing to the transverse relaxation. We speculate that this
additional dephasing process is driven by intra-molecular instantaneous diffusion due to the electron spin flip-flop processes
resulting from simultaneous excitation of $\left|-\frac{1}{2}\right\rangle \rightarrow \left|\frac{1}{2}\right\rangle$ transitions belonging to both $Gd^{3+}$ ions of one Gd-ruler. This seems to
be consistent with the $T_m$ values derived from the fits (see Table-S2) which show identical slow parts for both samples as one
would expect for the same matrix and different fast parts as a result of two Gd-rulers with different dipolar couplings, due to
different Gd-Gd distances, and therefore different spin flip-flop rates. The inter-molecular instantaneous diffusion process is
less effective at concentrations as low as used in this study and is therefore considered to not contribute.





Inversion recovery data shown in Fig. S2b have been well fitted with the sum of two mono-exponential functions with fast and slow components (see Table-S1). In addition, we provided a single $T_1$ value that was determined from a mono-exponential function fit to the inversion recovery data. It is interesting to note that the longer $T_m$ component is comparable to the shorter $T_1$ component.

For DEER experiments, the weak coupling approximation is generally expected to hold when the PO offset is
significantly larger than the dipolar coupling between the coupled spins. This is usually fulfilled for $Gd^{3+}$-$Gd^{3+}$ distance measurements where a frequency offset of at least 100 MHz is often used. In both Gd-rulers, Gd-ruler (2.1nm) and Gd-ruler (6.0 nm), the expected dipolar couplings are 5.6 and 0.2 MHz respectively and are far below the smallest frequency offset of 120 MHz used in our measurements. However, in the present work, as well as in the literature, artefacts in the spectra are observed for distances below 3-4 nm (Raitsimring et al., 2007;Dalaloyan et al., 2015;Cohen et al., 2016;Manukovsky et al.,
2017). Such artefacts mainly manifest themselves as a damping of the dipolar modulations in the time domain, which in turn results in an artificial broadening of the distance distribution. This has previously been explained in terms of unwanted excitation of flip-flop transitions within the central line. For the highest sensitivity in $Gd^{3+}$-$Gd^{3+}$ DEER measurements, the pump pulse is usually set at the maximum of the ED-FS spectrum to ensure the deepest modulation depth (and the observer is often set just outside the central line). Under such conditions, the central transition, $\left|-\frac{1}{2}\right\rangle \to \left|\frac{1}{2}\right\rangle$, contributes most to the pumped
spins, whereas, just away from the central transition, the $\left|-\frac{3}{2}\right\rangle \to \left|-\frac{1}{2}\right\rangle$ transition becomes the more dominant contribution to the observer spins (see Fig. S4a,b). The DEER signal is thus the result of the difference between the energy levels associated with the two transitions $\left|-\frac{3}{2}(A),\frac{1}{2}(B)\right\rangle \to \left|-\frac{1}{2}(A),\frac{1}{2}(B)\right\rangle$ and $\left|-\frac{3}{2}(A),-\frac{1}{2}(B)\right\rangle \to \left|-\frac{1}{2}(A),-\frac{1}{2}(B)\right\rangle$. The associated energies of these two states are degenerate to first order of the ZFS and only a fairly small splitting is induced by the second order ZFS contribution and this falls within the range of the dipolar couplings corresponding to short distances between $Gd^{3+}$ ions.
Therefore, the weak coupling approximation is no longer satisfied and the secular pseudo-terms describing the flip-flop effects cannot be ignored. This has been confirmed theoretically and investigation has shown that the artefacts are only significant for short distances, where the dipolar coupling is large, and when either the pump or observer pulse is set on the $\left|-\frac{3}{2}\right\rangle \to \left|-\frac{1}{2}\right\rangle$ transition adjacent to the $\left|-\frac{1}{2}\right\rangle \to \left|\frac{1}{2}\right\rangle$ or vice versa (Cohen et al., 2016;Manukovsky et al., 2017). However, when other transitions are selected by the observer pulse, it was shown that these artefacts are strongly reduced (Manukovsky et al., 2017).
This was originally experimentally confirmed in experiments with a dual-mode cavity (Cohen et al., 2016), and is also clearly seen in the experiments described here. This is particularly demonstrated in Fig. 5 where the pump pulse is set on the $\left|-\frac{1}{2}\right\rangle \to \left|\frac{1}{2}\right\rangle$ transition and the observer pulse is moved progressively further away from the central transition, which gradually reduces the contribution of the adjacent $\left|-\frac{3}{2}\right\rangle \to \left|-\frac{1}{2}\right\rangle$ transition (see Figs. S5).



For the short Gd-ruler (2.1 nm) clearer modulations and narrower distance distributions are observed as the frequency
offset is increased. Clearly visible modulations in the time domain are observed at the largest PO offset of 900 MHz (see Fig.
5a), although simulations have indicated that some residual effects from the pseudo-secular term can be observed even at such
PO offsets (Manukovsky et al., 2017). Interestingly, small artefacts are even observed for the longer Gd-ruler (6.0 nm) in Fig.
3 where better fits to the expected Pake pattern are obtained at the larger 420 MHz frequency offset and the related distance
distribution has its peak at the expected 6.0 nm distance (see Fig. 4b).

We note that in all the DEER studies reported so far, the central $\left|-\frac{1}{2}\right\rangle \rightarrow \left|\frac{1}{2}\right\rangle$ transition has always been selected, either
for the pump or the observer pulse. It had generally been assumed that there would be too big a sensitivity penalty to do
otherwise, and the advantage of operating at high fields was mainly associated with narrowing the line and achieving a higher
degree of excitation of the central transition. This led to the view that it is necessary to choose a Gd spin label with a large ZFS
when measuring short distances to reduce the effect of unwanted flip flops (Dalaloyan et al., 2015).

In this present work, we also investigated the DEER set-up where the pump and observer pulses are placed on either
side of the central transition, thus avoiding any excitation of the central transition completely (see Fig. S6a). These DEER
experiments provide time domain data shown in Fig. 7a with clear oscillations smoothly damped to the limit value (modulation
depth), giving well-defined Pake patterns and narrow distance distributions that are strikingly similar to those obtained for
structurally related nitroxide-rulers with comparable spin-spin distances (Godt et al., 2006;Jeschke et al., 2010). This is
evidence that when the central transition does not play a role in the $Gd^{3+}$- $Gd^{3+}$ DEER measurements, the mixing of states has
no major effect, as they do not share energy levels with those involved in the pump and observer transitions.

This suggests that Gd systems with lower ZFS are to be preferred (see Table-S1), because it is then easier to avoid
the central transition. In this case we would also expect the amplitude of other transitions to increase (per unit bandwidth), and
relaxation effects, due to thermally assisted fluctuations in the ZFS, to reduce (Raitsimring et al., 2014), which will further
increase detection sensitivity.

Table 1 indicates the highest echo SNR was obtained for Gd-ruler (2.1 nm), when neither pump nor observer are
placed at the central transition. Modulation depth, although reduced, was still reasonable, and thus we still obtain excellent
overall sensitivity under this condition. Note, Gd-ruler (2.1 nm) would be expected to have approximately 10 times higher
echo SNR compared to Gd-ruler (6.0 nm), just from the shorter time window required for the DEER measurement. In the
measurements presented here, this increase in sensitivity is only partially offset by the increased bandwidth required, and the
reduced power available from the EIK / isolator / switch combination at the band edges of the EIK amplifier. The available
power as a function of frequency is shown in Fig. S1b.

The ability to measure accurately short distances with Gd-based spin labels has implications for the choice of spin
label location in a biological compound, particularly for in-cell measurements. Biological samples are usually highly
protonated, which results in shorter phase memory times than for the model compounds used in this study. This means that the



relative gain in sensitivity from measuring a short 2.1 nm distance compared to a long 6.0 nm distance will be much larger compared to the cases considered here.

Measurements at low concentration levels also reduce the background contribution to the DEER traces that come from excitation of other spins. This is particularly important for $Gd^{3+}$- $Gd^{3+}$ DEER measurements where modulation depths are low, as small errors in background correction can make a significant contribution to uncertainties in the calculated distance distribution. In these experiments, raw DEER traces (see Figure S3b) from 40 μM Gd-ruler samples, i.e. 80 μM in $Gd^{3+}$, required relatively small background corrections compared to the modulation depth. However, a much larger background correction was found to be required at somewhat higher spin label concentrations (Dalaloyan et al., 2015).

The obtained sensitivity indicates that high quality measurements should be available at much lower concentration levels. There is also scope for further improvement by increasing both the shot repetition rate and averaging times and operating at lower temperatures, when the central transition is not excited, as well as using different spin labels. Backshort positions were also optimised for cross-polar isolation rather than matching out the echo signal, which can make a difference of a factor of 2 in sensitivity. Other groups have demonstrated a significant sensitivity benefit from the use of broadband chirped pulses in DEER measurements on $Gd^{3+}$ systems (Bahrenberg et al., 2017;Doll et al., 2015). These methodologies particularly lend

themselves to high power wideband systems like HiPER and promise significant further gains.

Even allowing for the shorter relaxation times commonly observed with spin labelled biological samples these results suggest that it is technically feasible to obtain high quality spectra for $Gd^{3+}$-$Gd^{3+}$spin label DEER measurements at sub-μM concentrations. We have also observed promising results with $Gd^{3+}$ spin labelled biological samples and we will report on these in a future publication.


**5 Conclusion**

In the present work we have investigated two Gd-rulers, with Gd-Gd distances of 2.1 and 6.0 nm, using a $Gd^{3+}$ complex with a moderate ZFS of 1060 MHz. We have performed a variety of $Gd^{3+}$-$Gd^{3+}$ DEER measurements with different offsets between pump and observer pulses, using a non-resonant induction mode cavity. This is a flexible wideband measurement set-up with

relatively easy sample handling, where excellent signal-to-noise is observed.

We have shown that, in agreement with previous experimental results, larger PO offsets significantly reduce the artefacts that are commonly observed for Gd-Gd distances below 3-4 nm, but also even appear to be of benefit in the case of larger distances, such as 6.0 nm.

More importantly we have shown significantly improved distance distributions at short distances by completely

avoiding excitation of the central transition in the DEER experiment, $\left|-\frac{1}{2}\right\rangle \rightarrow \left|\frac{1}{2}\right\rangle$, and mostly selecting $\left|-\frac{7}{2}\right\rangle \rightarrow \left|-\frac{5}{2}\right\rangle$, $\left|-\frac{5}{2}\right\rangle \rightarrow \left|-\frac{3}{2}\right\rangle$ and $\left|-\frac{3}{2}\right\rangle \rightarrow \left|-\frac{1}{2}\right\rangle$ transitions. This methodology still gives high signal-to-noise (per unit measurement time) while obtaining much improved fitting to expected Pake patterns. This is a strong motivation to select and/or develop Gd-based spin labels with as small a ZFS as possible, and measure using wideband spectrometers at moderately high magnetic



fields where the central transition narrows as field increases. The sensitivity is already high but we envisage considerable scope
for improvement.

**Code and data availability**

The research data supporting this publication can be accessed at DOI: https://doi.org/10.17630/96ab76ee-38f4-468f-9ea8-e947f638261f


**Author contributions**

MQ and AG designed and synthesised the Gd-rulers. HEM, RIH, PASC and GMS designed and built HiPER. HEM, RIH, JEL, AF and GMS devised the experiments which were performed by HEM, RIH and MJT. HEM and GMS chiefly wrote the manuscript, with further input from all authors.


**Competing interests**

The authors declare that they have no conflict of interest.

**Acknowledgements**

It is a pleasure to acknowledge both Dr Duncan Robertson (University of St Andrews) for useful discussions on hardware, and Professor Daniella Goldfarb (Weizmann Institute of Science) for useful discussions on Gd-based spin labels. HEM, GMS, JEL, RIH, PASC, and MJT are part of StAnD, which is a major collaboration between EPR groups at St Andrews and Dundee Universities.

**Financial support**

We would like to acknowledge EPSRC (EP/R)13705/1) for current funding on the HiPER project, and the Wellcome Trust for a multi-user equipment grant (099149/Z/12/Z) for upgrades on the Q-band system. We thank the Royal Society for an International Exchanges Grant and The Weizmann-UK Joint Research Program for allowing bilateral travel and research between the University of St Andrews and the Weizmann Institute of Science. JEL thanks the Royal Society for a University
Research Fellowship. MJT thanks EPSRC for a CM-CDT studentship (EP/LO15110/1). MQ and AG thank the Deutsche Forschungsgemeinschaft (DFG) for funding within SPP 1601 (GO555/6-2).

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
