# Peer review of "High sensitivity Gd3+- Gd3+ EPR distance measurements that eliminate artefacts seen at short distances"

_Magnetic Resonance, 2020_

## Referee Comment (RC1) · Gunnar Jeschke (Referee) · 28 Aug 2020

The authors demonstrate convincingly that by using a high-power W-band spectrometer with an only weakly resonant shorted waveguide end instead of a microwave resonator, the problem of level-mixing related perturbation of large dipolar frequencies can be avoided. This problem makes extraction of distance distributions between Gd(III) labels precarious at distances shorter than 3 nm. Hence, the new approach solves an important problem for broader application of Gd(III)-Gd(III) distance measurements. Furthermore, the authors show convincingly that it can be advantageous to avoid excitation of the central transition of Gd(III) altogether in DEER experiments, which comes

as a surprise. The manuscript is mostly clear and concise, the data is of high quality, and its analysis is mostly adequate. However, there are a few minor problems that require revision.

General:

1. The text refers mostly to frequency offsets when discussing different excitation schemes, but the Figures use PnOn codes. Please give frequency offsets directly in figures.

2. The claim on similar data quality as with nitroxide labels would have been much more convincing at intermediate distances, where nitroxide labels can convincingly resolve an asymmetry of the distance distribution that is related to flexibility of the ruler backbone. For that, the 2.1 nm may be too short and the 6 nm slightly too long (at the maximum dipolar evolution time achieved here).

3. section 65: Gd(III)-Gd(III) RIDME was first demonstrated and the overtone problem noticed in 2014 (DOI: 10.1021/jz502129t)

4. section 70: "applying the wormlike chain model". I do not find this in either the results section, discussion, or Supplementary Material.

5. section 155: Why did you measure Tm, T1 only at the maximum of the Gd(III) spectrum? These relaxation times are known to differ between central and satellite transitions and you focus on satellite transitions. Please make at least a remark that differences are to be expected and cite literature for that.

6. section 205: For systems other than Gd(III), Tm also can rarely be fitted with a single exponential function. Please rephrase to avoid the impression that this is a peculiar feature of Gd(III).

7. section 210: Why did you use fixed exponents 1 and 2 for fitting? I do not see a good theoretical reason for that. Fit quality is hard to ascertain in Fig. S2a (please use a representation as in Figure 3a, black versus red line), However, looking closely I
am not convinced that it is good. Please check if the fit residual is reasonably close to white noise.

8. Figure 3a: The fits for P1O1 and P2O2 do not compare favorably to the one shown by Dalaloyan et al. for the same compound (DOI: 10.1039/c5cp02602d). Do you have any idea why that may be the case?

9. section 265: "We do not expect any orientation selection and so the Pake pattern spectra reported in Fig. 6a show strong distortions". This sentence is unclear. Please rephrase.

10. section 310: I recommend more cautious wording regarding the origin of the second component, as you do not have a comparison with a sample with only a single Gd(III).

11. Figure S3: How can you have a rising background for P3O3? This needs to be commented.

---

## Referee Comment (RC2) · Stefan Stoll (Referee) · 9 Sep 2020

This manuscript describes DEER distance measurements on rigid Gd-Gd rulers in a high-power W-band spectrometer with a weakly resonant probe. Excellent sensitivity is demonstrated. It is shown that short-distance artifacts due to dipolar state mixing are suppressed by using a large pump-observe separation and by avoiding the central transition.

The work is well executed. The manuscript is well written. It provides novel and important insights. I recommend publication, after the following comments are addressed.

1. Some of DEER experiments are performed outside the central transition, but Tm and T1 values are reported only for the central transition. What are Tm and T1 for the pump and observer positions on the non-central transitions?

2. Line 111: Is it possible to give G/W^1/2 conversion efficiencies for the shorted waveguide used in this work, and for a standard cylindrical cavity as reference?

3. Line 272: How significant do the authors think are the differences between the data obtained at 840 MHz and 900 MHz offset? Are they within or outside the expected run-to-run scatter of the experiment?

4. Line 245: A pump-observe offset of 900 MHz is mentioned for the 6 nm ruler, but the data show 120 and 420 MHz offset only (Fig.3,4,S3a,S4).

5. Figs.4 and 6: What do the shaded areas in a) and b) indicate?

6. Fig.5d and 7b: What does the black arrow indicate?

7. Fig.S3a: What is the reason the background in the P3O3 measurement is rising, as opposed to decaying?

8. Table S3: What is T1 in the last column? Footnote 1 is not clear.

9. Table S1: Separate last column into two, one with the linewidths, and with references.

10. Line 313ff: I don't quite understand the author's arguments concerning intra-molecular instantaneous diffusion contribution to dephasing. The modulation depth is only a few percent, so only a few percent of spins get excited by each pulse. Simultaneous excitation of both spins within the same molecule therefore has very low probability. Some clarification would be useful.

11. Line 391ff: What are "backshort positions", and what does it mean to "match out the echo signal"?

12. Line 397: Claiming that sub-$\mu$M concentrations are technically feasible is a bit overly speculative. That would correspond to a ca. 50-fold reduction in concentration compared to the presented data, and a 2500-fold extension of the measurement time, for example from 1 hour to 3 months. Doubling the repetition rate to 6 kHz (more is not feasible given the T1) shortens this to 1.5 months, still not feasible. I suggest removing the statement about sub-$\mu$M concentrations.

---

## Referee Comment (RC3) · Alberto Collauto (Referee) · 27 Sep 2020

The authors propose a very interesting application of a high-bandwidth, high-power W-band setup to measure undistorted Gd(III)-Gd(III) DEER traces even for short distances, condition under which the mixing of the $|+\frac{1}{2},-\frac{1}{2}\rangle$ and $|-\frac{1}{2},+\frac{1}{2}\rangle$ states caused by the pseudosecular terms of the dipolar Hamiltonian results, under normal measuring conditions, in dampening of the dipolar modulation. A very interesting conclusion is proposed suggesting the use of the already available Gd(III)-based tags with low zero-field splitting even for short distances, provided that both the pump pulse and the detection sequence completely avoid the excitation of the central transition.

[Figure]

The manuscript is well written, and definitely suitable for publication on Magnetic Resonance; the conclusions are substantial and nicely supported by the presented data and analyses. However, there are some points that I would like to be addressed by the authors.

1. The style of the references is not homogeneous: in some cases the full DOI hyperlink is reported, whereas in other ones only the DOI number is displayed; some references make use of journal abbreviations, whereas in other ones the full journal title is mentioned. Besides, the absence of spacing and/or indentation makes it really hard to find a specific item. Moreover, references having the same first author are not always listed chronologically. I advise to follow thoroughly the author guidelines.

2. As far as I could see, no specific literature for Gd(III) labelling of DNAs has been cited although reference has been made in the text to this application (line 49).

3. I found the nomenclature proposed in Table 1 rather unclear; for example, why is a 10 ns-long pump pulse set to the maximum of the central transition once identified as P1 and once identified as P3 (6.0 nm Gd ruler)? I would find easier for the reader to have the relevant experimental conditions reported for each experiment (pulse length and frequency offset) in the figure caption or as inset, and, to improve the readability of the manuscript, I would consider moving the sensitivity considerations reported in Table 1 to the supporting material.

4. Is the (rather lengthy) discussion about the echo decay traces relevant for the purpose of this paper? After all, the measurements were performed on the maximum of the central transition, whereas the DEER detection sequence was always placed at spectral positions where the largest contribution to the echo comes from other transitions. A possible solution could be to move this section to the supporting material.

5. A high sensitivity of the experimental setup is claimed. However, a rather large sample amount (around 75 $\mu$L of a 40 $\mu$M solution, hence 3 nmol) was used compared to the typical ones used for conventional W-band or Q-band spectroscopy (around 5 $\mu$L

of a 40 $\mu$M solution, hence 0.2 nmol; 15 times smaller!), or even X-band spectroscopy (around 20 $\mu$L of a 40 $\mu$M solution, hence 0.8 nmol). An extension of the proposed approach to applications where the limiting factor is the sample amount, such as investigations inside cells or on systems that are challenging to express and/or label, is therefore in my opinion still not straightforward.

6. Throughout the main paper and the SI plots belonging to the same figure have different sizes and are not always aligned (see for example Figures 3 a/b, 5, S3, S4, S5).

7. Table 1: the shot repetition time should be given in time units; what is reported is the shot repetition rate.

8. Table 1: why was the shot repetition rate decreased from 3 kHz to 1 kHz for some of the measurements on the 2.1 nm Gd ruler (see Table 1)? Are measurements available to justify this choice?

9. Table 1: what was the used value of $\tau$1 for the DEER experiments?

10. Were the DEER measurements performed with or without a phase cycling of the $\pi$/2 pulse? If without, which precautions were taken so as to have no constant offset of the DEER traces?

11. In Figure S3a the intermolecular contribution for the experimental condition P3O3 has been modelled as an increasing function, a clearly unphysical assumption (as also stated by the authors). The analysis of these experimental data has to be repeated by taking an exponential decaying function. Furthermore, the primary data are displayed only for t >= 0; is this the way in which the data were recorded? If so, why? If not, it would be advisable to plot the whole data, in such a way that the maxima of the recorded traces are visible.

12. Figures 5d and 7b: what do the black arrows highlight?

13. Table S1: which distribution of E values was taken to fit the experimental data

shown in Figure 2? Were the simulations perform assuming a monomodal distribution of D values around +D or a bimodal distribution of D values around ±D? (I am not able to deduce this information from lines 197-199 of the main text).

14. Table S2: is the time corresponding to a decay of the echo signal to 10% of its initial value given as $\tau$ or $2\tau$? In which units is this value reported?

15. Captions of the Tables S2/S3: what is x? Was the dead time $2\tau 0$ taken into account for the fit of the echo decay curves? (This is relevant as the traces were fitted with a non-exponential function).

16. Table S3: a biexponential behavior of the inversion recovery curves has been reported. Were other kind of experiments attempted aimed at minimizing the role of spectral diffusion? Besides, a T1 value resulting from a monoexponential fit of the experimental traces has been reported but no comparison between the biexponential and monoexponential fits is shown in Figure S2.

17. Figure S2: because of the poor resolution I can hardly see the experimental data points.

18. Figure S2a: what was the minimum used value of $\tau$? This can't be deduced from the figure, where the first point of the decay trace has been set at $2\tau = 0$.

19. Figure S2b: the inversion recovery curves have not been collected till a plateau corresponding to the full recovery of the echo signal has been observed. This may result in severe uncertainties in the estimation of the longitudinal relaxation rate by fitting of the experimental data (Table S3).

20. Caption of Table S2 and lines 312-313 of the main text: why the fit of the echo decay curves has been described as a "sum of two stretched exponential functions" although for one of the components the exponent has been fixed to 1?

21. Figure S3: given the amount of free space on the page, I would consider useful to quickly recap, maybe in the form of a table, the relevant settings corresponding to the

different traces.

22. Figures S4, S5, S6: in my opinion, a reminder to the legend of Figure 2 for what concerns the color code used in the simulation of the EDFS-EPR spectra would be useful.

23. In my opinion, it would be useful to add the frequency response of the EIKA, which dominates the bandwidth of the system, to the plots in the supporting materials showing the excitation profiles of the pump and detection pulses. This would make immediately clear to the reader where the pulses have been positioned within the bandwidth of the transmission chain.

24. Figures 3, 5, 7, S4, S5, S6: how was the excitation profile of the detection sequence calculated?

---

## Author Comment (AC1) · 14 Oct 2020

**Reply to Comments by Prof Guunar Jeschke**

**The authors demonstrate convincingly that by using a high-power W-band spectrometer with an only weakly resonant shorted waveguide end instead of a microwave resonator, the problem of level-mixing related perturbation of large dipolar frequencies can be avoided. This problem makes extraction of distance distributions between Gd(III) labels precarious at distances shorter than 3 nm. Hence, the new approach solves an important problem for broader application of Gd(III)-Gd(III) distance measurements. Furthermore, the authors show convincingly that it can be advantageous to avoid excitation of the central transition of Gd(III) altogether in DEER experiments, which comes as a surprise. The manuscript is mostly clear and concise, the data is of high quality, and its analysis is mostly adequate. However, there are a few minor problems that require revision.**

We would like to thank Prof Jeschke for his kind words and careful reading of the manuscript and his helpful comments and suggestions.

**General:**

1. **The text refers mostly to frequency offsets when discussing different excitation schemes, but the Figures use PnOn codes. Please give frequency offsets directly in figures.**

We are keen to keep the PnOn codes as they are used in all the tables and the text, and this scheme was only chosen after examining a number of alternatives. However, we have now added frequency offsets to all the figures.

2. **The claim on similar data quality as with nitroxide labels would have been much more convincing at intermediate distances, where nitroxide labels can convincingly resolve an asymmetry of the distance distribution that is related to flexibility of the ruler backbone. For that, the 2.1 nm may be too short and the 6 nm slightly too long (at the maximum dipolar evolution time achieved here).**

This is a very good suggestion, and (co-author) Prof Godt also specifically suggested we check for this asymmetry. Unfortunately, we do not observe this asymmetry in either ruler, very possibly for the reasons the referee suggests. However, we agree that this will be a good check at intermediate distances and we will add a statement to that effect.

3. **section 65: Gd(III)-Gd(III) RIDME was first demonstrated and the overtone problem noticed in 2014 (DOI: 10.1021/jz502129t)**

This was an unfortunate oversight and we have added this reference.

4. **section 70: "applying the wormlike chain model". I do not find this in either the results section, discussion, or Supplementary Material.**

We agree the intended reference (Dalaloyan 2015) for this statement was not clear and we have now added a sentence to clarify.

5. **section 155: Why did you measure Tm, T1 only at the maximum of the Gd(III) spectrum? These relaxation times are known to differ between central and satellite transitions and you focus on satellite transitions. Please make at least a remark that differences are to be expected and cite literature for that.**

We completely agree measuring $T_M$ and $T_1$ at the offset frequencies used would have been both helpful and highly appropriate. It was an initial oversight. Unfortunately, immediately following the set of experiments described in the paper, the spectrometer was effectively rebuilt to incorporate a wideband AWG. The lab was then locked down due to Covid and we have not since been in a position to remeasure. We have measured $T_M$ in the same system as a function of offset at Q-band and as expected it shows $T_M$ becoming very significantly shorter with offset in line with results originally reported by Raitsmiring. We also now report in the SI, Q-band measurements with both pump and probe at offset frequencies (on one side of the central resonance). We cannot rule out that the substantial gain in sensitivity observed at W-band compared to Q-band is partly due to a difference in relaxation rates. We would point out that the excellent S/N observed at W-band is despite the expected reduction in $T_M$.

We will make it clear that $T_M$ was not measured at offset frequencies, and include a statement that we would expect $T_M$ to be transition dependent for this model system along with relevant reference(s). We will indicate the change in $T_M$ expected based on Q-band measurements.

6. **section 205: For systems other than Gd(III), Tm also can rarely be fitted with a single exponential function. Please rephrase to avoid the impression that this is a peculiar feature of Gd(III).**

We agree this statement was not clear and we will rephrase.

**7. Section 210: Why did you use fixed exponents 1 and 2 for fitting? I do not see a good theoretical reason for that. Fit quality is hard to ascertain in Fig. S2a (please use a representation as in Figure 3a, black versus red line), However, looking closely I am not convinced that it is good. Please check if the fit residual is reasonably close to white noise.**

The fit is actually extremely good ($R^2 = 0.9999$). If we look at the residual, we see a tiny bit of ESEEM at the very start, and then (at very small amplitude but well above the noise) we see a modulation that corresponds to the electron dipolar coupling. The residual is close to white noise at the end of the trace after the dipolar oscillation has decayed.

If we fit to the sum of two exponentials with different exponents, and let the exponents be free parameters, we get best fits where the exponents are found to be extremely close to 2 and 1. There is theoretical work by Salikhov that indicates that when Wt >>1 then the exponent is expected to be 1 and when Wt <<1 then you expect the exponent to be equal to 2 (where W is the rate of the dephasing process). We think it reasonable to suggest that the exponent 2 component is attributable to nuclear dipolar coupling and it is not unreasonable to believe there could be a very fast component. However, on reflection we agree that we should be more cautious in attributing a specific mechanism to this component.

The original basis for our speculation was we previously found (in unpublished results) that this same double exponent fit also offers excellent fits for large multi-spin systems with nitroxide spin labels (independent of sample concentration, but where the fast fluctuation depends on excitation bandwidth). But we haven't yet done similar experiments with Gd.We will use a similar representation as Fig 3a to make the data points clearer and give statistical

measures to indicate the fit is actually very good. We will add comments indicating why exponents 1 and 2 were chosen.

8. **Figure 3a: The fits for P1O1 and P2O2 do not compare favorably to the one shown by Dalaloyan et al. for the same compound (DOI: 10.1039/c5cp02602d). Do you have any idea why that may be the case?**

We would actually claim that the experimental data and fit for 6 nm-ruler are almost identical to the equivalent experimental data and fit of Dalaloyan at W-band (allowing for differences in modulation depth and S/N). It maybe the referee, was actually looking at the experimental data and fit for measurements made at Q-band, given in the same paper, for the same system. This does give a different modulation and, visually, a better fit. It also gives a different distance to that found at W-band. Previous computational modelling in Manukovsky 2017, that took into account pseudosecular terms, had shown that small distortions are still expected even at 5 nm distances, under experimental conditions comparable to P101, P202 (and for the Q-band data shown in Dalaoyan). One thus might expect Q-band data to be more susceptible to pseudosecular effects as the central transition is broader, and that has been our experience for other measurements with Gd spin-labelled systems (not shown). So we are more confident about the W-band data.

9. **section 265: "We do not expect any orientation selection and so the Pake pattern spectra reported in Fig. 6a show strong distortions". This sentence is unclear. Please rephrase.**

We were trying to convey that the distortions observed are not expected to be due to orientation selection (as might be the case for nitroxides). But we completely agree the current phrasing is unclear and we will change it.

10. **section 310: I recommend more cautious wording regarding the origin of the second component, as you do not have a comparison with a sample with only a single Gd(III).**

The current phrasing ("we speculate") was meant to convey a degree of caution, but as discussed above, we will change the wording so we do not attribute a specific mechanism to the second component.

11. **Figure S3: How can you have a rising background for P3O3? This needs to be commented.**

We had decided to include this background to be consistent with the methodology that was used in determining the background for all the other traces, together with a a comment in the caption stating this was unphysical. It is not completely clear why the background appears slightly different for this trace, compared to others. However, we agree that it casts unnecessary doubt on the results and we have now changed the fit to a slightly falling background.
We would stress this leads to almost exactly the same distance and distance distribution, but a slightly less perfect Pake pattern. We thus feel it does not change the underlying argument. We would also stress that the background in all the measurements is extremely low relative to previous measurements made on the same system (e.g. Dalolayan 2015). This reduces uncertainties in determining the distribution. The discrepancy probably arises because the time trace was not long enough to accurately determine the background as the oscillations have not completely died out. We will add explanatory comments.

---

## Author Response (AR1)

Below we give our (now slightly modified) responses to referees, indicating where we have made modifications to the text in responses to referee comments. We believe we have addressed all the comments, although in one or two cases we have chosen not to make a change, believing the already published response was sufficient. We hope we have justified this in the text. We would like to thank all referees, as we believe all their comments have made for a better paper.

We did discover one numerical error in our previous published response to referees when we quoted a sensitivity improvement of 72 with respect to a measurement made at Q-band (now included in the SI). It is actually 24. This number and comparison was not mentioned in the original manuscript, and the correct number is included in the new manuscript but we would be keen to add a correction to our responses.

We then give a marked-up script showing where the paper has been changed. There are also a few very minor extra changes made in the uploaded document, not shown here, but these are all very minor typographical changes.

Graham and Hassane

Prof Graham Smith
Dr Hassane El Mkami

**Reply to Comments from Gunnar Jeschke**

**The authors demonstrate convincingly that by using a high-power W-band spectrometer with an only weakly resonant shorted waveguide end instead of a microwave resonator, the problem of level-mixing related perturbation of large dipolar frequencies can be avoided. This problem makes extraction of distance distributions between Gd(III) labels precarious at distances shorter than 3 nm. Hence, the new approach solves an important problem for broader application of Gd(III)-Gd(III) distance measurements. Furthermore, the authors show convincingly that it can be advantageous to avoid excitation of the central transition of Gd(III) altogether in DEER experiments, which comes as a surprise. The manuscript is mostly clear and concise, the data is of high quality, and its analysis is mostly adequate. However, there are a few minor problems that require revision.**

We would like to thank Prof Jeschke for his kind words and careful reading of the manuscript and his helpful comments and suggestions.

**General:**

1. **The text refers mostly to frequency offsets when discussing different excitation schemes, but the Figures use PnOn codes. Please give frequency offsets directly in figures.**

We are keen to keep the PnOn codes as they are used in all the tables and the text, and this scheme was chosen after examining a number of alternatives. However, we have now added frequency offsets to all the figures.

2. **The claim on similar data quality as with nitroxide labels would have been much more convincing at intermediate distances, where nitroxide labels can convincingly resolve an asymmetry of the distance distribution that is related to flexibility of the ruler backbone. For that, the 2.1 nm may be too short and the 6 nm slightly too long (at the maximum dipolar evolution time achieved here).**

This is a very good suggestion, and (co-author) Prof Godt also specifically suggested we check for this asymmetry. These is some evidence for asymmetry Gd-ruler (6.0 nm), but we do not see this at the shorter distance very possibly for the reasons the referee suggests. We have added a statement to that effect.

3. **section 65: Gd(III)-Gd(III) RIDME was first demonstrated and the overtone problem noticed in 2014 (DOI: 10.1021/jz502129t)**

This was an unfortunate oversight and we have added this reference.

4. **section 70: "applying the wormlike chain model". I do not find this in either the results section, discussion, or Supplementary Material.**

We agree the intended reference (Dalaloyan 2015) to this statement was not clear and we have now added a sentence to clarify.

5. **section 155: Why did you measure Tm, T1 only at the maximum of the Gd(III) spectrum? These relaxation times are known to differ between central and satellite transitions and you focus on satellite transitions. Please make at least a remark that differences are to be expected and cite literature for that.**

We completely agree measuring $T_M$ and $T_1$ at the offset frequencies used would have been both helpful and highly appropriate. It was an initial oversight.  Unfortunately, immediately following the set of experiments described in the paper, the spectrometer was effectively rebuilt to incorporate a wideband AWG.  The lab was then locked down due to Covid and we have not since been in a position to remeasure. We have measured $T_M$ in the same system as a function of offset at Q-band and as expected it shows $T_M$ becoming very significantly shorter with offset in line with results originally reported by Raitsmiring. We also now report in the SI, Q-band measurements with both pump and probe at offset frequencies (on one side of the central resonance).  We cannot rule out that the substantial gain in sensitivity observed at W-band compared to Q-band is partly due to a difference in relaxation rates. We would point out that the excellent S/N observed at W-band is despite the expected  reduction in $T_M$.

   We have made it clear that $T_M$ was not measured at offset frequencies, and included a statement that we would expect $T_M$ to be transition dependent for this model system along with relevant reference(s).

6.   **section 205: For systems other than Gd(III), Tm also can rarely be fitted with a single exponential function. Please rephrase to avoid the impression that this is a peculiar feature of Gd(III).**

We agree this statement was not clear and we will rephrase.

**7. Section 210: Why did you use fixed exponents 1 and 2 for fitting? I do not see a good theoretical reason for that. Fit quality is hard to ascertain in Fig. S2a (please use a representation as in Figure 3a, black versus red line), However, looking closely I am not convinced that it is good. Please check if the fit residual is reasonably close to white noise.**

The fit is actually extremely good ($R^2 = 0.9999$).  If we look at the residual, we see a tiny bit of ESEEM at the very start, and then (at very small amplitude but well above the noise) we see a modulation that corresponds to the electron dipolar coupling. The residual is close to white noise at the end of the trace after the dipolar oscillation has decayed.

If we fit to the sum of two exponentials with different exponents, and let the exponents be free parameters, we get best fits where the exponents are found to be extremely close to 2 and 1.  There is work by Salikhov that suggests that when Wt $\gg$1 then the exponent is expected to be 1 and when Wt $\ll$1 then you expect the exponent to be equal to 2 (where W is the rate of the dephasing process).  We think it reasonable to suggest that the exponent 2 component is attributable to nuclear dipolar coupling and it is not unreasonable to believe there could be a very fast component. However, on reflection we agree that we should be more cautious in attributing a specific mechanism to this component.

The basis for our speculation is we previously found (in unpublished results) that this same double exponent fit also offers excellent fits for large multi-spin systems with nitroxide spin labels (independent of sample concentration, but where the fast fluctuation depends on excitation bandwidth) and also provides a much better fit than a single stretched exponential. But we haven't yet done similar experiments with Gd.

We have now used a similar representation as Fig 3a to make the data points clearer and have given statistical measures to indicate the fit is actually very good. We have also added comments indicating why exponents 1 and 2 were chosen.

8. **Figure 3a: The fits for P1O1 and P2O2 do not compare favorably to the one shown by Dalaloyan et al. for the same compound (DOI: 10.1039/c5cp02602d). Do you have any idea why that may be the case?**

We would actually claim that the experimental data and fit for 6 nm-ruler are almost identical to the equivalent experimental data and fit of Dalaloyan at W-band (allowing for differences in modulation depth and S/N). It maybe the referee, was actually looking at the experimental data and fit for measurements made at Q-band, given in the same paper, for the same system. This does give a different modulation and, visually, a better fit. It also gives a different distance to that found at W-band. Previous computational modelling in Manukovsky 2017, that took into account pseudosecular terms, had shown that small distortions are still expected even at 5 nm distances, under experimental conditions comparable to P101, P202 (and for the Q-band data shown in Dalaoyan). One thus might expect Q-band data to be more susceptible to pseudosecular effects as the central transition is broader, and that has been our experience for other measurements with Gd spin-labelled systems (not shown). So we are more confident about the W-band data.

9. **section 265: "We do not expect any orientation selection and so the Pake pattern spectra reported in Fig. 6a show strong distortions". This sentence is unclear. Please rephrase.**

We were trying to convey that the distortions observed are not expected to be due to orientation selection (as might be the case for nitroxides). But we completely agree the current phrasing is unclear and we have changed it.

10. **section 310: I recommend more cautious wording regarding the origin of the second component, as you do not have a comparison with a sample with only a single Gd(III).**

The current phrasing ("we speculate") was meant to convey a degree of caution, but as discussed above, we have changed the wording to indicate that more evidence is required.

11. **Figure S3: How can you have a rising background for P3O3? This needs to be commented.**

We had decided to include this background to be consistent with the methodology that was used in determining the background for all the other traces, together with a a comment in the caption stating this was unphysical. It is not completely clear why the background appears slightly different for this trace, compared to others. However, we agree that it casts unnecessary doubt on the results and we have now changed the fit to a slightly falling background.
We would stress this leads to almost exactly the same distance and distance distribution, but a slightly less perfect Pake pattern. We thus feel it does not change the underlying argument. The discrepancy probably arises because the time trace was not long enough to accurately determine the background as the oscillations have not completely died out.

We now just show a fit with a falling background with associated distance distribution and Pake pattern.

**Reply to Comments from Stefan Stoll**

**This manuscript describes DEER distance measurements on rigid Gd-Gd rulers in a high-power W-band spectrometer with a weakly resonant probe. Excellent sensitivity is demonstrated. It is shown that short-distance artifacts due to dipolar state mixing are suppressed by using a large pump-observe separation and by avoiding the central transition.**

**The work is well executed. The manuscript is well written. It provides novel and important insights. I recommend publication, after the following comments are addressed.**

We would like to thank Prof Stoll for his careful reading of the manuscript and many insightful comments, which we address below.

1. **Some of DEER experiments are performed outside the central transition, but Tm and T1 values are reported only for the central transition. What are Tm and T1 for the pump and observer positions on the non-central transitions?**

   We completely agree it would have been useful and appropriate to measure Tm away from the central transition at the offsets used. At the time of the experiments it was an oversight. Unfortunately immediately after these experiments the spectrometer was rebuilt to incorporate a wideband AWG and then the lab was locked down when COVID hit and so at the moment it is not possible to incorporate this data (at W-band). We have Q-band data and we will refer to this and other references. We would point out that $T_M$ is expected to get shorter away from the central transition but we still achieve excellent S/N despite this.

2. **Line 111: Is it possible to give G/W^1/2 conversion efficiencies for the shorted waveguide used in this work, and for a standard cylindrical cavity as reference?**

   It is possible to quote an effective conversion efficiency based on the typical length of the $\pi/2$ pulse (6 ns for a S=1/2 sample) in the waveguide, although of course the $B_1$ field varies significantly across the sample. For a fair comparison one should really quote this based on the power incident on the sample holder, which is not always clear when comparing systems, and of course it depends on the chosen bandwidth of the standard cavity. This type of comparison thus requires many caveats, which we are not keen to enter into in this paper. However, in the interests of discussion, a reasonable estimate in our system might be to assume 625 W at the sample, giving c ~ 0.6 $G/W^{1/2}$. This conversion efficiency is comparable to an X-band commercial cavity (optimised for concentration sensitivity) used in pulsed operation and with a comparable sample volume and comparable kW input power. As sensitivity scales with $\omega_0^2$, very substantial sensitivity gains become possible as long as linewidth does not get very significantly broader. A critically coupled W-band cylindrical cavity might have a conversion efficiency that is 15 x larger than the waveguide sample-holder here (but with a much smaller sample volume and bandwidth).

   We have given a conservative estimate for c in the text.

3. **Line 272: How significant do the authors think are the differences between the data obtained at 840 MHz and 900 MHz offset? Are they within or outside the expected run-to-run scatter of the experiment?**

We think there are small differences, but we agree they are not large, and they were included partly as we had the data sets. The point of including measurements with seemingly similar offsets is that at 840 MHz and 900 MHz offset we are very close to the maximum bandwidth available from the EIK, where the power output starts to significantly degrade at band edges., so we are testing the bandwidth limits.

4. **Line 245: A pump-observe offset of 900 MHz is mentioned for the 6 nm ruler, but the data show 120 and 420 MHz offset only (Fig.3,4,S3a,S4).**

   Many thanks – we have corrected this typo.

5. **Figs.4 and 6: What do the shaded areas in a) and b) indicate?**

   It is essentially a guide to the eye, but we have added notes in the caption explaining

6. **Fig.5d and 7b: What does the black arrow indicate?**

The arrow indicates 94 GHz, the nominal centre frequency of our W-band EIK amplifier, which has a bandwidth of just less than 1 GHz. We have added an explanation in the captions to make this clear.

7. **Fig.S3a: What is the reason the background in the P3O3 measurement is rising, as opposed to decaying?**

The underlying reason is that the oscillations have not decayed fully by the end of the time trace and so it is difficult to determine accurately. We chose to show this background (with a note it was not physical) to be consistent about the way we determined backgrounds for all the other traces. i.e. by optimising the resulting Pake pattern.

We have now changed this to give a slightly decaying background. This now gives a marginally worse Pake pattern, but essentially exactly the same distance and distance distribution.

8. **Table S3: What is T1 in the last column? Footnote 1 is not clear.**

We agree and have changed the footnote.

9. **Table S1: Separate last column into two, one with the linewidths, and with references.**

We have done this.

10. **Line 313ff: I don't quite understand the author's arguments concerning intra- molecular instantaneous diffusion contribution to dephasing. The modulation depth is only a few percent, so only a few percent of spins get excited by each pulse. Simultaneous excitation of both spins within the same molecule therefore has very low probability. Some clarification would be useful.**

As per the discussion with Prof Jeschke we have attempted to be more cautious with this statement, but we would point out that we are measuring at the central transition and the presence of pseudosecular interactions shows that we cannot treat it as a simple dilute system.

11. **Line 391ff: What are "backshort positions", and what does it mean to "match out the echo signal"?**

In common microwave terminology a backshort is a short circuit termination in a waveguide whose position can be adjusted with respect to some reference plane. The reflection from the top of the sample and this termination can create a weak resonant circuit, which can significantly enhance the magnitude of the cross-polar signal. Thus sensitivity (echo signal) is thus maximised for certain positions of the backshort. We have added lines of explanation. In the text

12. **Line 397: Claiming that sub-μM concentrations are technically feasible is a bit overly speculative. That would correspond to a ca. 50-fold reduction in concentration compared to the presented data, and a 2500-fold extension of the measurement time, for example from 1 hour to 3 months. Doubling the repetition rate to 6 kHz (more is not feasible given the T1) shortens this to 1.5 months, still not feasible. I suggest removing the statement about sub-μM concentrations.**

The 1.5 month (or 3 month) time-scale suggested for a measurement at sub-μM concentrations would be correct if we needed to maintain the same S/N to extract useful information from the spectra. But the echo S/N after ~ 1.5 hours for 2.1 nm ruler is 8300. Even with only 2.1 % modulation, one would still have acceptable S/N if we reduced S/N by a factor of 10 reducing averaging time by a factor of 100, bringing averaging times < 1 day for sub-uM concentrations. Many published measurements are made with this averaging time. So we thus stand by our statement that sub-μM concentrations are feasible (right now).

To further emphasize this point we also now include additional comparative Q-band measurements in the SI. In one example, where both pump and probe are offset from the central transition, if we compare to our P3O data for 2.1 nm sample, we have a lower (echo) sensitivity by a factor of ~ 8 and a lower modulation depth by a factor of 3, leading to an effective reduction in S/N of 24. Satisfactory S/N was still obtained by averaging for approximately one day. We would suggest that S/N would still be acceptable with somewhat lower averaging times.

In the discussion, we also point out there are realistic ways to further improve sensitivity. Higher sensitivity would be expected with Gd-complexes with smaller zero-field splittings. There is scope to achieve higher sensitivity by using a wideband AWG to increase both pump and probe excitation bandwidths. Although not discussed in the paper, we also believe there is also scope to improve the conversion factor of the sample holders, whilst maintaining all their other advantages. We thus believe the sub-μM claim will ultimately prove to be relatively conservative.

We have added a sentence or two to make these points clearer, and have included Q-band data in the SI.

**Reply to Comments from Alberto Collauto**

**The authors propose a very interesting application of a high-bandwidth, high-power W-band setup to measure undistorted Gd(III)-Gd(III) DEER traces even for short distances, condition under which the mixing of the (…) states caused by the pseudo-secular terms of the dipolar Hamiltonian results, under normal measuring conditions, in dampening of the dipolar modulation. A very interesting conclusion is proposed suggesting the use of the already available Gd(III)-based tags with low zero-field splitting even for short distances, provided that both the pump pulse and the detection sequence completely avoid the excitation of the central transition.**

**The manuscript is well written, and definitely suitable for publication on Magnetic Resonance; the conclusions are substantial and nicely supported by the presented data and analyses. However, there are some points that I would like to be addressed by the authors.**

We thank Dr Collauto for his kind words above, and careful reading of the manuscript. We reply to his helpful and interesting comments below.

1. **The style of the references is not homogeneous: in some cases the full DOI hyperlink is reported, whereas in other ones only the DOI number is displayed; some references make use of journal abbreviations, whereas in other ones the full journal title is mentioned. Besides, the absence of spacing and/or indentation makes it really hard to find a specific item. Moreover, references having the same first author are not always listed chronologically. I advise to follow thoroughly the author guidelines.**

This appears to have been a problem with ENDNOTE. We believe we have now corrected any inconsistencies in the referencing.

2. **As far as I could see, no specific literature for Gd(III) labelling of DNAs has been cited although reference has been made in the text to this application (line 49).**

We have now added a reference.

3. **I found the nomenclature proposed in Table 1 rather unclear; for example, why is a 10 ns-long pump pulse set to the maximum of the central transition once identified as P1 and once identified as P3 (6.0 nm Gd ruler)? I would find easier for the reader to have the relevant experimental conditions reported for each experiment (pulse length and frequency offset) in the figure caption or as inset, and, to improve the readability of the manuscript, I would consider moving the sensitivity considerations reported in Table 1 to the supporting material.**

We have used such nomenclature P1 and P3 to emphasize that these two positions, although having similar pulse lengths, and positioned at the central transition, are at different frequencies. Note that the observer frequency is kept at 94 GHz and the pump frequency is varied so there are difference. We chose this naming scheme only after considering many alternatives.

With regard to the sensitivity, this is mentioned in the title and high concentration sensitivity is emphasized in the abstract, and we are not aware of any experimental results that show a higher concentration sensitivity for these systems. So we feel it is an important part of the

manuscript. These numbers allow other groups to directly compare sensitivity. The paper is not just about measuring short distances by having probe and pump away from the central transition. It is the fact that one can still make the measurement with very high sensitivity that we feel makes the result useful and interesting.

4. **Is the (rather lengthy) discussion about the echo decay traces relevant for the purpose of this paper? After all, the measurements were performed on the maximum of the central transition, whereas the DEER detection sequence was always placed at spectral positions where the largest contribution to the echo comes from other transitions. A possible solution could be to move this section to the supporting material.**

We would claim that relaxation times and thus discussion of echo decay traces is highly relevant to sensitivity with regards to the practicality and design of potential experiments made at low sample concentrations. We completely agree it would be better to give the relaxation times at offset frequencies. At the time this was an oversight. Unfortunately, immediately after the experiments the spectrometer was rebuilt to incorporate a wideband AWG and then we had the lab lock-down, and it has not been possible to make these measurements since.

5. **A high sensitivity of the experimental setup is claimed. However, a rather large sample amount (around 75 µL of a 40 µM solution, hence 3 nmol) was used compared to the typical ones used for conventional W-band or Q-band spectroscopy (around 5 µL of a 40 µM solution, hence 0.2 nmol; 15 times smaller!), or even X-band spectroscopy (around 20 µL of a 40 µM solution, hence 0.8 nmol). An extension of the proposed approach to applications where the limiting factor is the sample amount, such as investigations inside cells or on systems that are challenging to express and/or label, is therefore in my opinion still not straightforward.**

We can only agree that it would be nice to have both extremely high concentration sensitivity and very little sample. However, this comment does not appear to take into account the significant loss of concentration sensitivity for small volume cavities, especially at lower frequencies. If you are not sample limited then (with some caveats) maximising sample volume, at a given frequency, will always give a larger signal. We have now attached data in the SI where we show measurements taken at Q-band at high power (150 W), using Bruker's large volume Q-band cavity (with comparable sample volume to that used here – 50-60 µL). For 2.1 nm, with both pump and probe on one side of the central transition, the concentration sensitivity is reduced by around a factor of 72, compared, to the W-band measurement corresponding to P3O in the paper. One might expect the concentration sensitivity of the small volume Q-band resonator (quoted) to be down a further factor of 4. The concentration sensitivity of the X-band resonators quoted are likely to be very significantly worse. Of course, for systems that are difficult to express, having 50 uL sample volumes is not necessarily trivial – but as discussed in the paper there are potentially relatively straightforward ways to further improve sensitivity (and hence potentially reduce volume and improve absolute sensitivity) and still reach sub-µM sensitivity. So we believe this to be a very promising and flexible approach. That is not to say there aren't other promising approaches, like the W-band resonator approach taken by our collaborators at the Weizmann Institute. But we believe it will be very challenging to significantly improve the sensitivity at X-band and Q-band to make them competitive, both in terms of absolute and concentration sensitivity for these types of samples.

We have added some more lines in the discussion, discussing sensitivity.

6. **Throughout the main paper and the SI plots belonging to the same figure have different sizes and are not always aligned (see for example Figures 3 a/b, 5, S3, S4, S5).**

This has now been significantly improved.

**Table 1: the shot repetition time should be given in time units; what is reported is the shot repetition rate.**

Many thanks – we have corrected this typo and changed to shot repetition rate.

7. **Table 1: why was the shot repetition rate decreased from 3 kHz to 1 kHz for some of the measurements on the 2.1 nm Gd ruler (see Table 1)? Are measurements available to justify this choice?**

The simple answer is that the 1 kHz measurements could have been made with a repetition rate of 3 kHz (or an even higher rate as Stefan Stoll suggests) and we would have had the same signal to noise in less time. At the time we were being very conservative in our choice. That is one of the reasons we give sensitivity measures to allow different results to be compared.  Results were not repeated, because immediately afterwards the experiments, the spectrometer front-end and detection system was rebuilt to incorporate a wideband AWG, which was a major change. We then had the COVID lab shut-down (and are still affected by it).

8. **Table 1: what was the used value of τ1 for the DEER experiments?**

The value is 300 ns and has been added to the figure caption.

9. **Were the DEER measurements performed with or without a phase cycling of the π/2 pulse? If without, which precautions were taken so as to have no constant offset of the DEER traces?**

We can measure with phase cycling, but these specific measurements were actually taken without phase cycling (for technical reasons).  Instead, offsets were removed by separate automatic measurements of the baseline, on either side of the echo.   This baseline was then subtracted (at the cost of a slight reduction in S/N).   We would add that offsets are known to be very low, and signals were relatively high in these experiments and so the correction was rather small.  We have added a line mentioning this.

10. **In Figure S3a the intermolecular contribution for the experimental condition P3O3 has been modelled as an increasing function, a clearly unphysical assumption (as also stated by the authors). The analysis of these experimental data has to be repeated by taking an exponential decaying function. Furthermore, the primary data are displayed only for t >= 0; is this the way in which the data were recorded? If so, why? If not, it would be advisable to plot the whole data, in such a way that the maxima of the recorded traces are visible.**

We chose to show the increasing intermolecular contribution (to be consistent with other results where contributions were chosen to give the best Pake pattern).  However, we agree that this just casts an unnecessary question mark on that result.  We have now fitted using a decaying contribution.  This leads to a slightly less optimum Pake pattern, but effectively exactly the same distance and distance distribution. The underlying problem is of course that

the oscillations have not died out by the end of the trace, and so it is difficult to fit the background with absolute confidence. It is not clear why this trace has a slightly different background. However, we would point out that the intermolecular contributions in all these traces are much smaller than have been observed before in experiments on these samples and thus potential errors in estimating distributions are correspondingly much smaller.

We chose to only show t > 0, as the traces are very long and thus the period where t < 0 is correspondingly very short relative to the total trace, and correspond to only a few points, with little extra information content.

Nevertheless, we have now added these points.

**12. Figures 5d and 7b: what do the black arrows highlight?**

The arrows indicate 94 GHz which is the centre frequency of the EIK. This was added to be a guide to the eye, to show how the pump and probe pulses were positioned relative to the centre frequency. We have added a note in the captions.

**13. Table S1: which distribution of E values was taken to fit the experimental data shown in Figure 2? Were the simulations perform assuming a monomodal distribution of D values around +D or a bimodal distribution of D values around ±D? (I am not able to deduce this information from lines 197-199 of the main text).**

Whilst some previous studies on Gd-spin labels samples have needed a bimodal distribution centred on +D and –D to simulate the observed spectra, in this study, we found we could get an excellent fit by using a monomodal distribution around D. We have added a note to make it clear how the Gd spectra was modelled

**14. Table S2: is the time corresponding to a decay of the echo signal to 10% of its initial value given as τ or 2τ? In which units is this value reported?**

This parameter is a function of 2τ, and the units are μs. We have clarified this in the text.

**15. Captions of the Tables S2/S3: what is x? Was the dead time 2τ 0 taken into account for the fit of the echo decay curves? (This is relevant as the traces were fitted with a non-exponential function).**

The x parameter corresponds to time and it has now been changed to t to make this clear. The dead time was taken into account in the fits. (However, we didn't observe a major difference to the fits, both with and without the dead time).

**16. Table S3: a bi-exponential behaviour of the inversion recovery curves has been reported. Were other kind of experiments attempted aimed at minimizing the role of spectral diffusion? Besides, a T1 value resulting from a mono-exponential fit of the experimental traces has been reported but no comparison between the biexponential and mono-exponential fits is shown in Figure S2.**

At 40 uM concentration we do expect much (intermolecular) spectral diffusion, although we cannot completely rule it out. We have added a statistical measure for the mono-exponential to give a measure for the quality of the fit.

**17. Figure S2: because of the poor resolution I can hardly see the experimental data points.**

That is partly because the fit is so good ($R^2$=0.9999), but we have now changed the way the experimental data is presented to hopefully make this clearer.

**18. Figure S2a: what was the minimum used value of $\tau$? This can't be deduced from the figure, where the first point of the decay trace has been set at $2\tau = 0$.**

The value of the interpulse $\tau$ was 300 ns and a note has been added to the figure caption.

**19. Figure S2b: the inversion recovery curves have not been collected till a plateau corresponding to the full recovery of the echo signal has been observed. This may result in severe uncertainties in the estimation of the longitudinal relaxation rate by fitting of the experimental data (Table S3).**

We agree that a slight error is possible, but the results are entirely consistent with Ref [Phys Chem Chem Phys, 18, 19037-19049] and were only used to estimate viable repetition rates. We have added a line of explanation.

**20. Caption of Table S2 and lines 312-313 of the main text: why the fit of the echo decay curves has been described as a "sum of two stretched exponential functions" although for one of the components the exponent has been fixed to 1?**

Many thanks. We agree the term "stretched" is confusing in this context and we have removed this.

**21. Figure S3: given the amount of free space on the page, I would consider useful to quickly recap, maybe in the form of a table, the relevant settings corresponding to the different traces.**

Tables have now been added in the available space showing the relevant settings corresponding to the different traces.

**22. Figures S4, S5, S6: in my opinion, a reminder to the legend of Figure 2 for what concerns the color code used in the simulation of the EDFS-EPR spectra would be useful.**

We have now added the legends, that were shown in Figure 2.

**23. In my opinion, it would be useful to add the frequency response of the EIKA, which dominates the bandwidth of the system, to the plots in the supporting materials showing the excitation profiles of the pump and detection pulses. This would make immediately clear to the reader where the pulses have been positioned within the bandwidth of the transmission chain.**

We have carefully considered this suggestion, but on balance we feel that this would make the resulting figures too cluttered. We have however pointed out in the caption that 94 GHz represents the centre frequency, and has been designated by a black arrow, where appropriate.

**24. Figures 3, 5, 7, S4, S5, S6: how was the excitation profile of the detection sequence calculated?**

An effective $B_1$ was calculated, derived from the length of the $\pi/2$ pulse that gave the largest signal, (allowing for the fact we are dealing with a high spin system). The predicted excitation profile for a refocussed echo was then calculated using simple spin mechanics, (Phys Chem Chem Phys., 2007, 1895-1910). The resulting excitation profile is narrower than the excitation profile of a simple $\pi$ pulse or a simple Hahn spin echo, as expected.

We have added a line and the reference.

[revised manuscript text omitted]

This is the subject of further investigation, but we predict that since the Gd-rulers investigated have a moderate ZFS (1060 MHz) (see **Table-S1**), the experimental results when avoiding the central transition suggests that Gd systems with lower ZFS are to be preferred because it is then easier to avoid the central transition. In this case we would also expect the amplitude of other transitions to increase (per unit bandwidth), and relaxation effects due to thermally assisted fluctuations in the ZFS to reduce (Raitsimring et al., 2014), which will further increase detection sensitivity.¶
At 40 µM molecular concentration,

We also note from the obtained sensitivity there is considerable scope to make high quality measurements at much lower concentration levels. …

[revised manuscript text omitted]